# Electrochemotherapy Intralesional Treatment in a Captive Peregrine Falcon (*Falco peregrinus*) with Dermal Squamous Cell Carcinoma

**DOI:** 10.3390/ani15070919

**Published:** 2025-03-22

**Authors:** Sara Peña, Otilia Ferrer, Beatriz Balañá, Mariana Lima, Laura Ordeix, Jaume Alomar

**Affiliations:** 1Department of Physology, Faculty of Medicine, University of Las Palmas de Gran Canaria (ULPGC), Campus Arucas, 35016 Las Palmas, Spain; 2Hospital Clínico Veterinario, University of Las Palmas de Gran Canaria, Campus Arucas, 35004 Las Palmas, Spain; otilia.ferrer@ulpgc.es; 3Medical and Surgery Unit, Deparment of Animal Pathology, Faculty of Veterinary Sciences, University of Las Palmas de Gran Canaria (ULPGC), Campus Arucas, 35004 Las Palmas, Spain; 4Unitat Quirúrgica Docent de Torrelameu Serveis Veterinaris, Universitat de Lleida, Torrelameu, 25138 Barcelona, Spain; beatrizbalanatapia@googlemail.com; 5Instituto de Ciências Biomédicas Abel Salazar, University of Porto, 4050-313 Porto, Portugal; mariana.cmql@gmail.com; 6Hospital Clínic Veterinari, Universitat Autònoma de Barcelona, Cerdanyola del Vallés, 08193 Barcelona, Spain; laura.ordeix@uab.cat; 7Servei de Diagnòstic de Patología Veterinària, Universitat Autònoma de Barcelona (UAB), Bellaterra, 08193 Barcelona, Spain; jaume.alomar@uab.cat

**Keywords:** oncology, electrochemotherapy, prey bird, avian oncology, cancer, chemotherapy, veterinary oncology

## Abstract

Birds of prey in captivity often live much longer than in the wild. However, they may also face a higher risk of developing tumors, such as squamous cell carcinoma (SCC). SCC is a common type of cancer in pet birds and has been reported in raptors like peregrine falcons, usually affecting the flank or thigh. Surgery is the main treatment, but it’s not always possible due to the tumor’s location or size. Electrochemotherapy (ECT), which improves drug delivery using electric pulses, is emerging as a promising option. This case report presents the successful use of ECT with bleomycin in a peregrine falcon with SCC, suggesting it could be a valuable treatment for cancer in birds of prey.

## 1. Introduction

Captive birds of prey consistently surpass the expected lifespan of their wild counterparts. For instance, falcons exhibit significant longevity differences depending on the species, with the American kestrel (*Falco sparverius*) averaging 2.3 years in the wild, while falcons in general may live between 15 and 20 years in captivity, with some individuals reaching up to 25 years [1]. Nevertheless, there is growing evidence to suggest that neoplastic processes are becoming more frequent within these species. Forbes et al. (2000) reported that the prevalence of neoplasms in raptors (order *Accipitriformes*, *Cathartiformes*, *Falconiformes*, and *Strigiformes*) may be greater than previously thought [2]. However, the current literature and research on neoplasms in birds of prey remain very limited, which makes an accurate assessment of the prevalence of tumors in these species challenging [2]. A retrospective study reported a 9.9% prevalence of neoplasia in *Strigiformes* as the basis of 131 postmortem cases, but no significant trends were observed in *Falconiformes* [3].

Squamous cell carcinoma (SCC) is a frequently observed neoplasm in avian species, predominantly pet birds [3]. With respect to birds of prey, few reports exist, although a retrospective study identified 13 cases, 7 of which involved *Falco peregrinus*. The most affected locations were the flank or thigh [2]. SCCs typically exhibit locally invasive behavior, but distant metastasis is uncommonly reported in birds. Although complete surgical excision is the optimal approach for achieving a complete response in SCCs [4], this is not always a viable option because of the extensive nature and invasiveness of some cases. It is therefore essential to evaluate the suitability of alternative treatment options. Systemic chemotherapy has been employed in avian species, but the specific protocols and doses remain largely empirically defined [5].

Electrochemotherapy is a therapeutic modality that employs electroporation to increase cellular membrane permeability without causing irreversible damage. This process facilitates the delivery of chemotherapeutic agents into tumor cells, thereby increasing their antineoplastic efficacy [6]. Electrochemotherapy with intralesional application of chemotherapeutic agent is a safer approach with fewer systemic risks than other treatment modalities. Bleomycin and cisplatin have been used as intralesional chemotherapies in avian species [7,8].

In veterinary oncology, electrochemotherapy has been successfully employed in various domestic and exotic species, including budgerigars (*Melopsittacus undulatus*) [7], cockatiels (*Nymphicus hollandicus*) [8], ferrets (*Mustela putorius furo*) [9], yellow-bellied sliders (*Trachemys scripta scripta*) [10], inland bearded dragons (*Pogona vitticeps*) [11], and four-toed hedgehogs (*Atelerix albiventris*) [12]. Given the lack of published data on electrochemotherapy in this species, this report aims to describe the treatment protocol used, providing a reference for future cases. The hypothesis is that electrochemotherapy with bleomycin is a safe therapeutic option for the management of squamous cell carcinoma in peregrine falcons.

## 2. Case Presentation

A female 7-year-old peregrine falcon (*Falconiformes*, *Falconidae*, *Falco peregrinus*) was presented for consultation due to the presence of two cutaneous lesions. The animal lived in a zoo, was bred in captivity, was fed fresh meat, frequently performed decoy flying exercises, and lived nearby with other birds of prey that did not have similar injuries. The lesions were located in the left axillary region and the right side of the coelomic cavity, extending laterally from the ribs and reaching the inguinal region. These lesions had been present for several months prior to presentation and had been treated with disinfection and topical antibacterial and antifungal therapies but did not respond. The animal had been wearing a harness for one year before the lesions appeared.

The animal exhibited an acceptable body condition (3/5) upon physical examination, with no visible weight loss. The animal also maintained a regular appetite. The bird displayed normal wing movement during both horizontal and vertical flight.

The skin surrounding both lesions was less densely feathered. Macroscopic observation of the lesion on the right side revealed a crusty surface with significant ulceration, leading to substantial tissue loss and disruption of tissue integrity, resulting in exposure of the underlying muscle tissue (Figure 1 and Figure 2). The lesion extended across an area of 1.5 cm^2^ and reached a depth of 1 cm.

Possible differential diagnoses for two chronic, ulcerative, and bleeding lesions in a peregrine falcon include traumatic ulceration and pressure sores due to chronic friction, foreign body reaction with secondary necrosis from prolonged contact with harness materials, and chronic bacterial dermatitis or deep pyoderma caused by pathogens such as *Staphylococcus* spp. or *Pseudomonas* spp. Additionally, mycotic dermatitis or deep fungal infection (e.g., *Aspergillus* spp., *Candida* spp.), squamous cell carcinoma (SCC) resulting from chronic irritation and UV exposure, or soft tissue sarcoma (e.g., fibrosarcoma) secondary to persistent inflammation should be considered. Other potential causes include ulcerative dermatitis associated with ectoparasites such as *Cnemidocoptes* spp. or feather mites, viral infections like *Avipoxvirus*, immune-mediated skin diseases such as pemphigus complex, or nutritional deficiencies, particularly hypovitaminosis A.

The lesion in the left axillary region was more superficial in nature and lacked the presence of ulceration. It exhibited the presence of multiple yellow-colored crusts, along with a central erosion area, and measured approximately 0.75 cm^2^ in longitudinal surface area (Figure 3 and Figure 4).

## 3. Materials and Methods

A complete blood count was performed, revealing no significant abnormalities in any of the parameters, with the exception of mild nonregenerative anemia. Bacterial and fungal cultures were also performed. *Pseudomonas* spp. and *Candida* spp. were isolated within the sample. Radiographs of the coelomic cavity were obtained to evaluate pulmonary structures and air sacs. No visible alterations were found.

An incisional biopsy was performed in order to obtain further information. Anesthesia was conducted via inhalation of isoflurane. The biopsy was processed routinely with paraffin-embedded tissue and hematoxylin–eosin (H&E) stain for histopathological study. Histopathological analysis revealed neoplastic proliferation, characterized by nonencapsulated, poorly demarcated, and infiltrative growth, which effaces and replaces the epidermis and dermis, reaching the surgical margins. The neoplasm was arranged in nests of multiple layers with squamous differentiation (keratin pearls), surrounded by desmoplastic mature connective tissue. The neoplastic cells were polygonal, with abundant eosinophilic cytoplasm and partially defined cellular borders. One or more nuclei were present, round to oval in shape, with stippled chromatin. The anisocytosis and anisokaryosis rates were high, and the mitotic count was moderate. Other microscopic findings included severe epidermal ulceration replaced by serocellular crusts, peripheral epidermal hyperplasia with hyperkeratotic orthokeratosis, and severe interstitial inflammatory infiltrate mainly composed of heterophils, lymphocytes, and plasma cells. The definitive diagnosis for both lesions was invasive squamous cell carcinoma, accompanied by ulcerative and heterophilic dermatitis. The nature of the dermatitis was determined to be subacute to chronic, with the presence of bacterial contamination (Figure 5).

In accordance with the initial culture antibiogram results, the patient was administered systemic treatment with injectable amikacin at a dosage of 15 mg/kg every 24 h over 3 weeks, in conjunction with local disinfection via 3% chlorhexidine and the application of a cream formulated from Asiatic spark (*Centella asiatica*). The secondary infection diminished with visible improvement, with a reduction in odor and the disappearance of the yellow-colored crusts.

Given the depth and extent of both lesions, surgical intervention poses a risk of compromising critical functions, such as wing movement, so other therapy modalities have been explored. Electrochemotherapy (ECT) with intratumoral chemotherapy was selected as the treatment modality. ECT has documented efficacy in managing cutaneous neoplasms in other species, through the enhancement of the uptake of chemotherapeutic agents, such as bleomycin and cisplatin, by permeabilizing electric pulses. This, in turn, leads to significant DNA damage and the induction of apoptosis in cancer cells.

Anesthesia was performed via inhalatory isoflurane (Vetflurane^®^ 1000 mg/g liquid for vapor inhalation, Virbac, Barcelona, Spain) with a mask, with an induction dose of 3% and maintenance at 1.5%. Meloxicam (Meloxidyl^®^ 5 mg/mL injectable solution for dogs and cats, Ceva Sante Animale, Barcelona, Spain) was administered subcutaneously at a dosage of 0.5 mg/kg every 24 h for a period of 10 d.

In total, two ECT sessions were performed, with a 90 d interval. Intratumoral bleomycin (Bleomicina Mylan^®^, 15.000 IU/10 mL, Prasfarma, S.L., Barcelona, Spain) was administered at a dose of 1500 IU/cm^3^. The dosage was calculated in accordance with the formula $A\times B\times C\times \frac{\pi }{6}$ (where *A*, *B*, and *C* represent the length, width, and thickness/height of the tumor, respectively) (Figure 6).

After injecting bleomycin intratumorally, sequences of eight biphasic pulses, each lasting 50 + 50 µsec, were delivered in bursts at 1000 V/cm via caliper electrodes. The procedure was repeated until all affected areas on both sides had been treated completely.

The pulses were generated via the Onkodisruptor Exp-Vet^®^ (Biopulse, Naples, Italy) electroporator with manually programmed plate electrodes. Ultrasound gel was applied to the treatment areas to facilitate the transmission of electrical impulses and enhance tissue permeability.

The electrical pulses were applied immediately after complete infiltration of the surface of the lesion with bleomycin, treating the lesion on the right side first, followed by the left side (Figure 7).

Buprenorphine (Buprecare^®^ 0.3 mg/mL, injectable solution, Divasa-Farmavic, S.A., Barcelona, Spain) at dose of 0.03 mg/kg every 12 h was administered for the management of pain and analgesia, in conjunction with meloxicam (Meloxidyl^®^ 5 mg/mL injectable solution for dogs and cats, Ceva Sante Animale, Barcelona, Spain) at dose of 0.5 mg/kg every 24 h for a period of two weeks. Daily cleaning of the wounds was carried out using saline solution. Imiquimod cream (Aldara^®^ 5% cream, Viatris Pharmaceuticals, S.L.U., Madrid, Spain) was applied three times per week, in combination with fusidic acid cream (Fucidine^®^ 20 mg/g, Laboratorios LEO Pharma S.A., Málaga, Spain).

After 90 days, a second session was performed using the automatic ElectroVet EZ^®^ (Leroy Biotech, Saint Orens de Gameville, France) electroporator with long needle electrodes (8 needles) at 1000 V/cm, with the same dosage calculation and protocol as in the first session. Repeat X-rays revealed no evidence of masses or potential metastases in the lungs or air sacs.

After the first ECT session, the lesion on the right side maintained dimensions of 1.5 cm^2^, but the depth was reduced to 0.1 cm, thus indicating a partial response (90%). The affected tissue underwent a process of regeneration at a deeper level, with muscle no longer being exposed. This outcome led to a reduction in deep tissue exposure and a decreased risk of secondary infections. The macroscopic appearance of the lesion revealed granulation tissue.

Conversely, the lesion on the left side exhibited signs of disease progression, particularly in the medial area of the left wing, where an initial accumulation of hyperplastic tissue and crusts was observed. This area developed an ulcer with a surface area of 0.8 cm^2^ (Figure 8 and Figure 9).

Following the second ECT treatment, the depth of the lesion on the right side further decreased, accompanied by an increase of 0.4 cm in surface width. Nevertheless, in contrast to the initial stage, no muscle exposure was present (Figure 10).

With respect to the left side, the lesion progressed at the thoracic/axillary level, with an ulcerative lesion measuring approximately 0.2 × 0.3 cm and a similar size to the medial lesion of the left wing. This progression was considered indicative of disease advancement in that area (Figure 11).

The survival time from December 2020 to March 2022 was 455 days, with an initial mild to partial response of the right lesion. The neoplasia remained stable in the right lesion but progressed slowly on the left side. It did not penetrate the muscle, in contrast to the initial lesion on the right side. No metastases were identified during necropsy.

Coelomic cavity radiographies did not identify any metastasis, and the tumor was classified as T3 according to the modified Owen classification used in veterinary oncology.

The falcon flap did not display any indications of adverse events. It is imperative to ensure effective pain and inflammation control, as well as the management of secondary infections, as observed in analogous cases across various species. The bird’s appetite remained unchanged, and periodic weight assessments demonstrated consistent results within the standard range for its species and sex.

Notably, throughout the duration of the treatment, the patient did not experience any difficulties in flying and was able to carry out natural behaviors, including hunting with a lure during both vertical and horizontal flights.

## 4. Discussion

Neoplastic disorders in birds are becoming more common than just a postmortem diagnosis due to an increasing body of knowledge and advancements in the field of avian medicine. The expectation of superior medical care demands accurate diagnosis, prognosis, and therapeutic options. However, published information regarding the prognosis and therapy of specific neoplasms remains limited in avian medicine [13].

In the context of captive raptors and birds of prey, the occurrence of neoplasms is not rare, underscoring the importance of comprehensive clinical evaluation and histopathological analysis for precise diagnosis [14].

The development of squamous cell carcinomas in avian species has been associated with chronic inflammatory conditions, such as chronic nonhealing wounds, chronic skin diseases, or areas subjected to repeated trauma [13]. This emphasizes the importance of conducting a biopsy, followed by histopathological analysis, in cases of persisting cutaneous lesions.

Although metastasis is uncommonly reported in avian squamous cell carcinomas, performing imaging exams to accurately stage the neoplasm is essential. Radiography is typically the primary screening modality for lung metastasis; however, computed tomography offers a more precise evaluation [15].

Electrochemotherapy has proven to be an effective therapeutic modality in the treatment of various neoplasias within human and veterinary medicine. The literature includes reports of squamous cell carcinoma [16,17], melanoma [18,19], soft tissue sarcomas [19,20], injection site sarcomas (ISSs) [21], mastocytomas [22,23], localized cutaneous lymphoma [24], plasma cell tumors [25], perianal tumors [26], and sarcoid lesions [27].

To the best of our knowledge, there have been no other documented cases of squamous cell carcinoma treated with electrochemotherapy in a peregrine falcon in the literature.

In the context of exotic animals, intralesional chemotherapy is regarded as a safer treatment modality, as it does not present systemic effects that are associated with intravenous chemotherapy. However, there is a paucity of research in this area, with no established dosage protocols, a paucity of described side effects, and a lack of standardization of sessions. All of the available research is extrapolated from the treatment of dogs and cats.

Electrochemotherapy allows increased penetration of the chemotherapeutic agent into tumor cells. Bleomycin is a glycopeptide antibiotic that is believed to function primarily by inhibiting DNA synthesis via the generation of free radicals, thereby inducing single and double-strand breaks in DNA [28]. Bleomycin exerts its effects during the process of cell division. Consequently, the onset of necrosis is evident only as cells begin to divide. The maximum effect is only observed following a period of several weeks [29].

Cisplatin is the chemotherapeutic agent with the most reports in the exotic medicine literature [30]. In this case, bleomycin was chosen because of some reports of superior results with this agent [8]. Therefore, a comparison of the use of these agents in the treatment of neoplasias in birds of prey would be relevant and interesting to evaluate.

ECT can be a valuable adjuvant therapy modality for locally invasive or incompletely excised neoplasms, providing local control of the tumors and improving the cosmetic appearance [7,8].

The management of SCC in raptors remains challenging due to limited standardized treatment protocols and the anatomical constraints that hinder complete surgical excision. As surgical removal remains the gold standard for SCC treatment, the outcomes vary depending on the feasibility of achieving clean margins. According to the clinical outcomes reported by Zehnder et al., 2018 [4], complete surgical excision was the most effective treatment, with 50% (4/8) of the cases achieving complete remission. However, recurrence or progression was observed in some cases, particularly when excision margins were insufficient. When combined with adjunctive therapies such as topical or nutraceutical treatments, systemic chemotherapy, or radiation therapy, surgical excision showed variable success rates, with one case achieving complete remission and another resulting in disease progression.

Non-surgical treatment options demonstrated mixed results. Systemic chemotherapy, when used alone, did not result in complete remission, with one case leading to progressive disease and another ending in euthanasia. When combined with cryotherapy, systemic chemotherapy resulted in partial remission in one case, while two other cases experienced disease progression. Strontium radiation therapy alone achieved a 20% (1/5) complete remission rate, but three of the treated cases resulted in euthanasia or death, indicating a poor overall prognosis [4].

Cryotherapy alone or in combination with other therapies yielded inconsistent outcomes. While one bird treated with cryotherapy alone showed stable disease, two others experienced disease progression. Intralesional chemotherapy combined with cryotherapy was associated with positive responses, achieving partial remission in the limited cases reported. External beam radiation, either alone or in combination with cryotherapy, demonstrated minimal efficacy, with only partial remission achieved in one case [4].

Compared to these conventional treatment modalities, ECT offers a minimally invasive alternative with localized drug delivery and enhanced cytotoxic effects. The case study on ECT treatment in a peregrine falcon demonstrated a significant reduction in tumor depth and stabilization of the disease, supporting its potential as an effective therapeutic approach. Unlike systemic chemotherapy, which has shown inconsistent outcomes in avian species, ECT minimizes systemic toxicity while enhancing drug uptake by tumor cells. The promising response observed in this case aligns with previous findings on ECT efficacy in other exotic and avian species.

Overall, the comparison of SCC treatments highlights the need for individualized therapeutic approaches, considering tumor location, invasiveness, and feasibility of complete excision. While surgical removal remains the preferred option, ECT represents a valuable alternative, particularly in cases where surgical intervention poses significant risks. Future studies should further explore the long-term efficacy of ECT in avian oncology, focusing on optimizing protocols and assessing recurrence rates.

In the present case, surgery was not a viable option because of the size and extent of the affected tissue. Consequently, an alternative approach was adopted with the objective of achieving local disease control and prolonging survival time. Electrochemotherapy successfully reduced the depth of the lesion, thereby facilitating secondary infection control. However, control of the neoplasia was not achieved at a superficial level, resulting in progressive, albeit always local, disease.

It is imperative to ensure a patient’s well-being; thus, the management of pain represents a vital component of the therapeutic regimen. ECT can result in moderate pain in the area of contact between the electrode and the skin. Pain in birds is often misinterpreted and underestimated. Birds present discrete signs of pain, underscoring the necessity for both practitioners and caretakers to be vigilant and aware of the signs exhibited throughout the treatment process. In this case, no signs of pain or discomfort were reported, with the raptor displaying normal activity and no alterations in appetite.

It can be argued that ECT is a potentially efficacious therapeutic modality for epithelial tumors in exotic animals, including birds and birds of prey. The treatment is associated with minimal secondary effects and allows for the maintenance of the disease and, in some cases, a reduction in the progression of the condition. In the present case, ECT allowed the deep closure of the lesion, thereby reducing the risk of infection and accidents in deep tissue areas. Additionally, this case establishes the foundation for a suggested treatment protocol in this species, previously undescribed, and opens the possibility of combining ECT with other therapeutic approaches, such as surgery.

Further research and studies are needed in these species to increase understanding of the impact and optimize the methodology, dosages, and scheduling of subsequent sessions.

## Figures and Tables

**Figure 1 animals-15-00919-f001:**
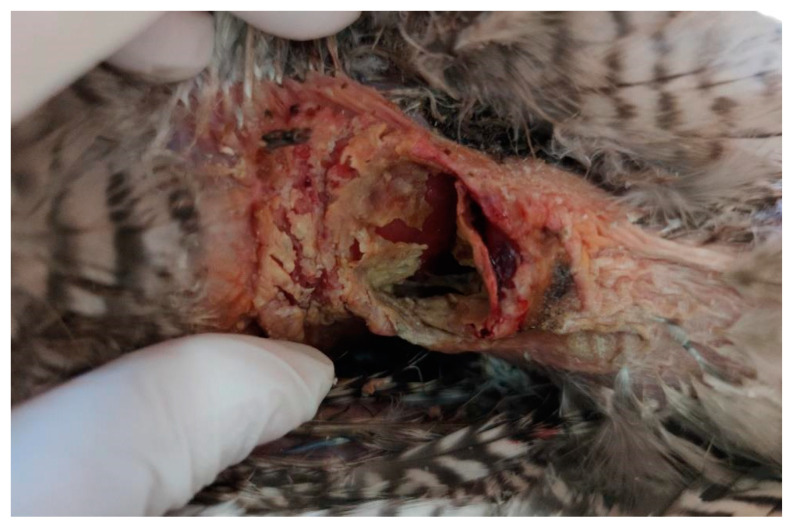
Right view of the lesion before cleaning the devitalized tissues and crust, showing a deeper wound.

**Figure 2 animals-15-00919-f002:**
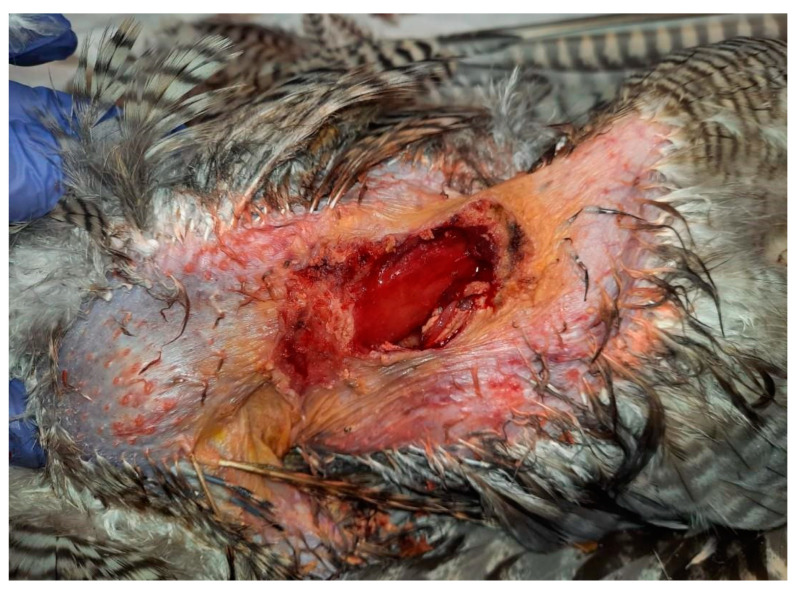
Right lesion characterized by deep ulceration with exposed muscle tissue after cleaning. Yellow-colored crusts are also visible, as well as feather loss.

**Figure 3 animals-15-00919-f003:**
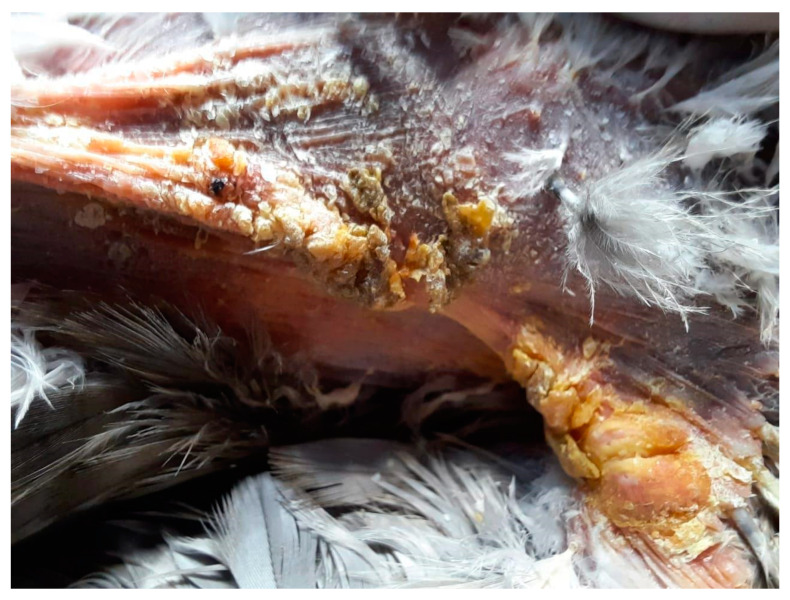
Left view of the axillar region with yellow crust and feather loss, before cleaning.

**Figure 4 animals-15-00919-f004:**
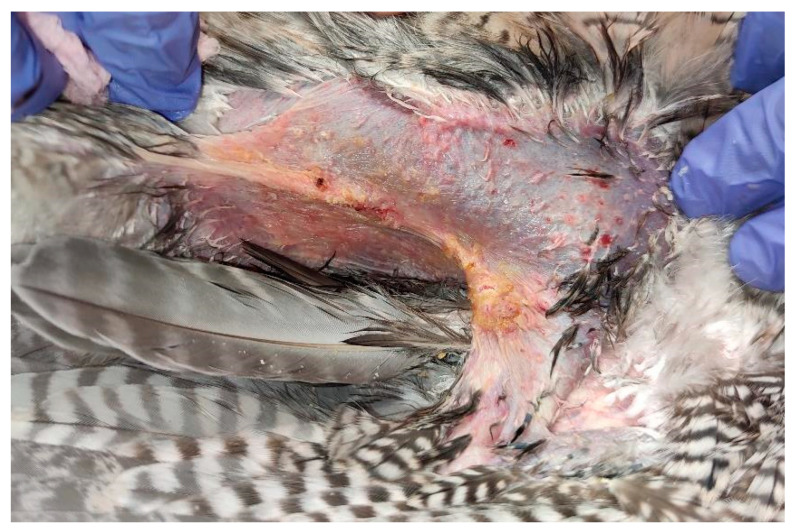
Left axillary’s lesion presented with erosion and yellow-colored crusts, after cleaning.

**Figure 5 animals-15-00919-f005:**
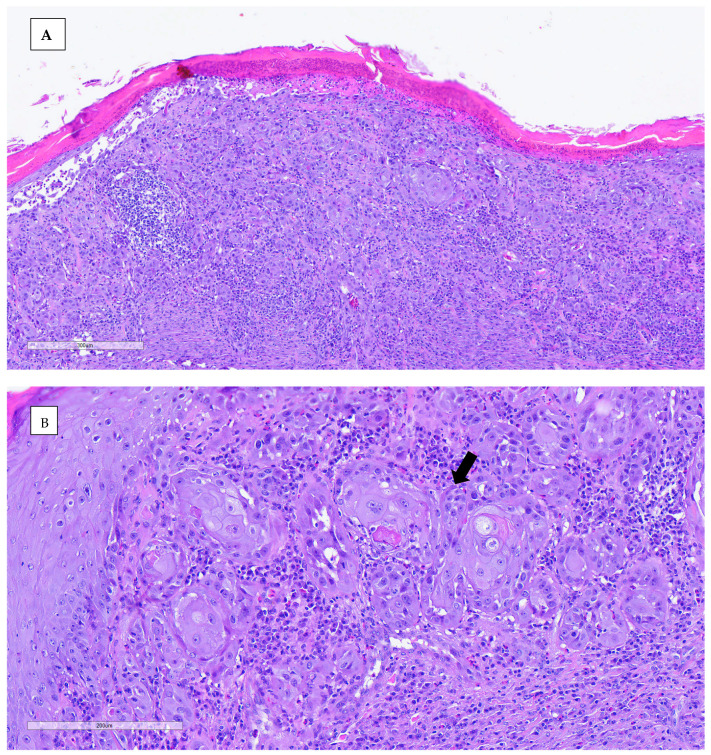
(**A**) The microscopic images showed orthokeratotic and epithelial neoplastic proliferation, not encapsulated or well delimited and infiltrative. Intense hyperkeratosis is observed in the non-ulcerated epidermis. (**B**,**C**) Neoplastic cells growing, forming nests of multiple layers with scaly differentiation, with some pearls of keratin observed. They are surrounded by an abundant amount of mature fibrous tissue interstitial (intense fibrous desmoplasia). The neoplastic cells are polygonal, with abundant eosinophilic cytoplasm and well-defined cytoplasmic borders.

**Figure 6 animals-15-00919-f006:**
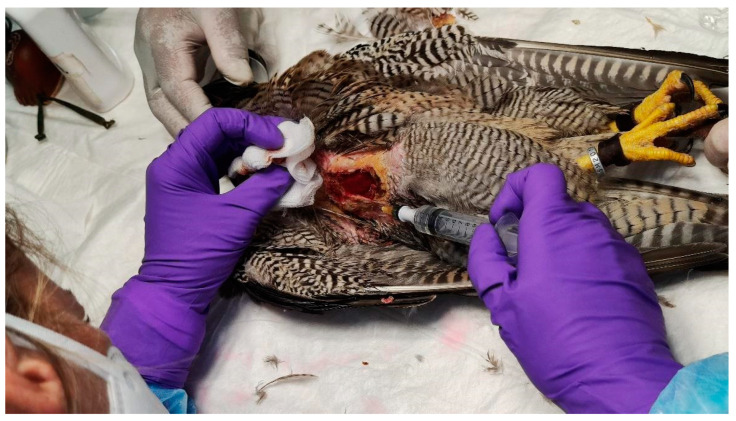
Application of intratumoral bleomycin to the edges and base of the neoplastic lesion.

**Figure 7 animals-15-00919-f007:**
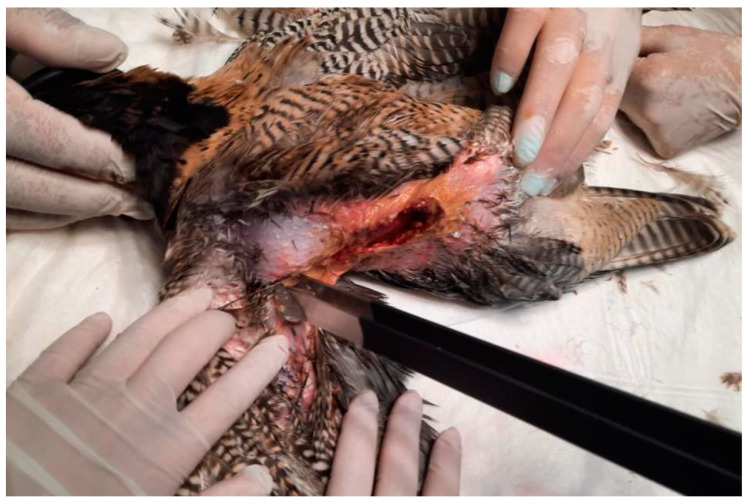
Application of electrical pulses using plate electrodes and ultrasound gel on the lesion located on the right side.

**Figure 8 animals-15-00919-f008:**
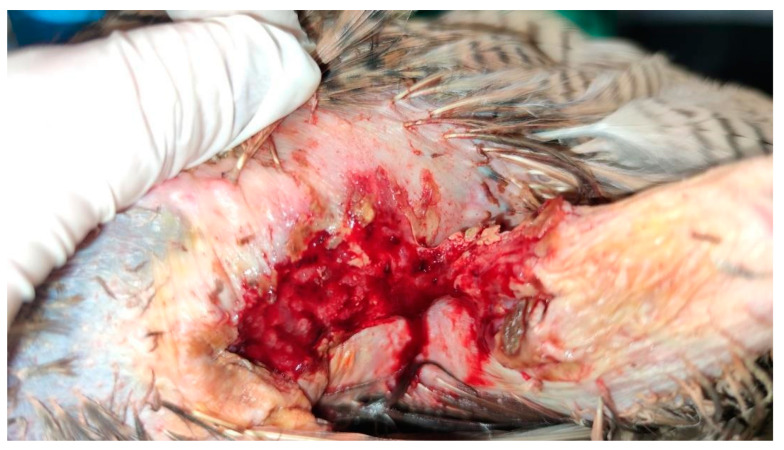
Right-side lesion prior to the second session of electrochemotherapy, showing a reduction in depth.

**Figure 9 animals-15-00919-f009:**
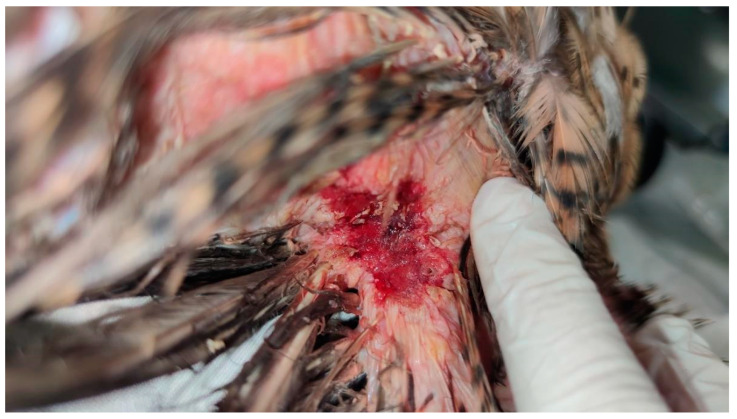
Left axilla lesion presenting an ulcer in the medial area.

**Figure 10 animals-15-00919-f010:**
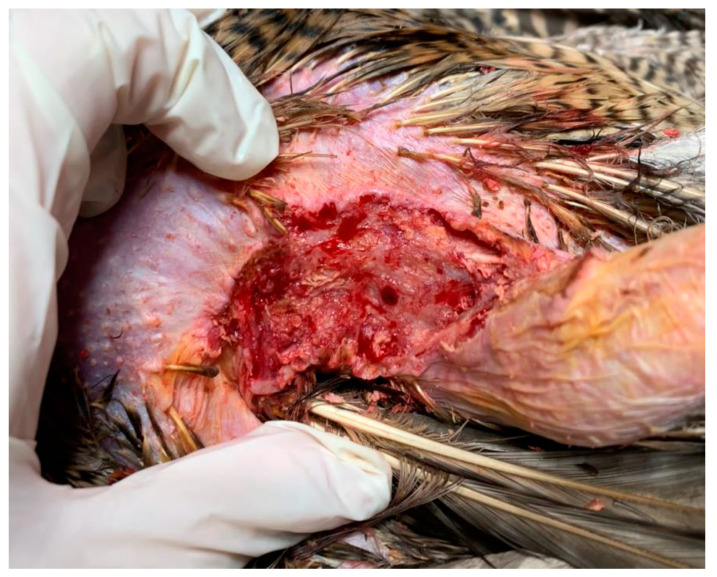
Right-side lesion after the second session of electrochemotherapy, showing no exposure of the muscle tissue but an increase in lesion length.

**Figure 11 animals-15-00919-f011:**
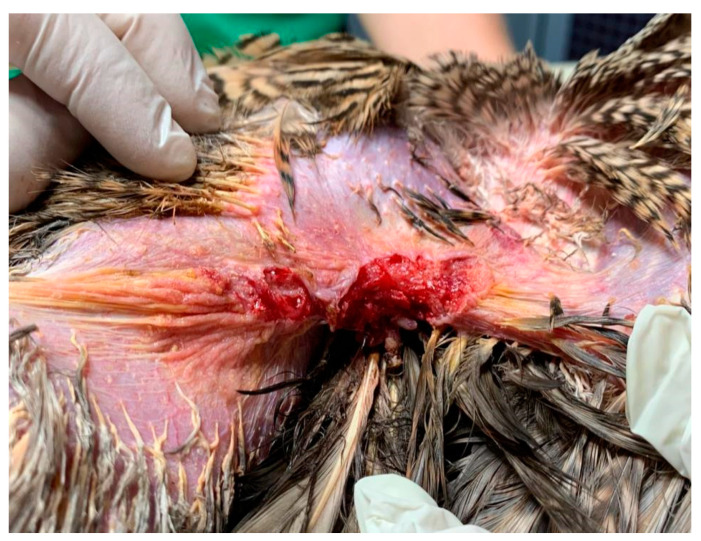
Left axillary lesion showing surface ulcers.

## Data Availability

The datasets used and/or analyzed during the current study are available from the corresponding author on reasonable request.

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
