# Peer review of "Electrochemotherapy Intralesional Treatment in a Captive Peregrine Falcon (*Falco peregrinus*) with Dermal Squamous Cell Carcinoma"

_animals, 2025, doi:10.3390/ani15070919_

Round 1

Reviewer 1 Report

Comments and Suggestions for Authors

Dear authors

Minor editing is suggested to be included here, as follows

TITLE

L2-L3 The title could be modified in order to highlight the chemotherapy treatment instead of the squamous cell carcinoma diagnosis in a captive peregrine falcon. An option of a potential new version of the title could be "Electrochemotherapy intralesional treatment in a captive peregrine falcon (Falco peregrinus) with dermal squamous cell carcinoma".

INTRODUCTION

L56-L65 This paragraph describing squamous cell carcinoma could be moved after the one of electrochemotherapy (L66-L72 and L73-L76). In order to characterize squamous cell carcinomas in birds, the authors are suggested to include information of both dermal, and oropharyngeal / pharyngoesophageal squamous cell carcinomas from the current literature in avian species.

Please use "yellow-bellied sliders" instead of "turtles" (L75); and "inland bearded dragon" instead of "reptiles" and add "four-toed" to "hedgehogs" (L76).

CASE PRESENTATION

L81 The scientific name of the peregrine falcon could be included as "(Falconiformes, Falconidae, Falco peregrinus)" instead of " ... (Falco peregrinus), belonging to the family Falconidae,..."

L81-L86 Please include more information about the patient such as housing, nutrition, prior treatments, etc. 

Comment 1. The inclusion of shortlisted tumoral and non-tumoral entities could be added as differential diagnosis for the dermal squamous cell carcinoma.

Comment 2. Figures. The reorganization of the figures in 3 plates is recommended. One plate can include the figures of the tumoral gross aspect before treatment. A second plate could include the pictures of the microscopic features of this tumoral entity. A third plate of the tumoral gross aspect after treatment can be included.

DISCUSSION

L240-L242 Please clarify is this tumoral entities belonged to human or veterinary medicine.

L244-L247 "These species ... albiventris) [11]" should be discard to avoid repetition with the information of the Introduction section.

L249-L250 Please avoid the inclusion of the scientific name "Falco peregrinus".

L270 to L291 Please include cited references to support several of the sentences included in these paragraphs.

Author Response

Comments L2-L3: The title could be modified in order to highlight the chemotherapy treatment instead of the squamous cell carcinoma diagnosis in a captive peregrine falcon. An option of a potential new version of the title could be "Electrochemotherapy intralesional treatment in a captive peregrine falcon (Falco peregrinus) with dermal squamous cell carcinoma".

Response: I totally agree. I just change.

Comments L56-L65: This paragraph describing squamous cell carcinoma could be moved after the one of electrochemotherapy (L66-L72 and L73-L76). In order to characterize squamous cell carcinomas in birds, the authors are suggested to include information of both dermal, and oropharyngeal / pharyngoesophageal squamous cell carcinomas from the current literature in avian species.

Response: I could change.

Comments: Please use "yellow-bellied sliders" instead of "turtles" (L75); and "inland bearded dragon" instead of "reptiles" and add "four-toed" to "hedgehogs" (L76).

Response: I just change it.

Comments L81 The scientific name of the peregrine falcon could be included as "(FalconiformesFalconidaeFalco peregrinus)" instead of " ... (Falco peregrinus), belonging to the family Falconidae,..."

Response: I just change it.

Comments L81-L86 Please include more information about the patient such as housing, nutrition, prior treatments, etc. 

Response: I just change it.

Comment 1. The inclusion of shortlisted tumoral and non-tumoral entities could be added as differential diagnosis for the dermal squamous cell carcinoma.

Response: I just change it.

Comment 2. Figures. The reorganization of the figures in 3 plates is recommended. One plate can include the figures of the tumoral gross aspect before treatment. A second plate could include the pictures of the microscopic features of this tumoral entity. A third plate of the tumoral gross aspect after treatment can be included.

Response: I could try it.

Comments DISCUSSIONL240-L242 Please clarify is this tumoral entities belonged to human or veterinary medicine.

Response: I just change it.

L244-L247 "These species ... albiventris) [11]" should be discard to avoid repetition with the information of the Introduction section.

Response: I just change it.

L249-L250 Please avoid the inclusion of the scientific name "Falco peregrinus".

Response: I just change it.

L270 to L291 Please include cited references to support several of the sentences included in these paragraphs.

Response: I just change it.

Reviewer 2 Report

Comments and Suggestions for Authors

Line 26 - I am not sure there is evidence for higher than expected prevalence - this is non-sensical.  you refer to Forbes line 48 who reports prevalence is greater than previously thought...what does this mean...greater than he thought or someone else - this is just an opinion - it would be a valid point if it was greater than a previous study....but greater than what someone thought is a pretty low bar in the scale of scientific evidence?

Line 48

I would agree with the other reviewers comments - have you any supporting evidence.  This should be available from the many breeding centres involved with breeding falcons commercially and also from the conservation breeding centres.  

line 81

more case information here please - sex, age (should be known if captive bred) and management (falconry or breeding bird) and diet if known.

line 88

what do you mean vertical flight - was this falcon still being flown and stooping even with such severe injuries that are pictured in figures 1 and 2.  

line 104

I find it surprising that there were no significant abnormalities - it almost seems inconcievable that the white cell count was not affected or out of normal ranges.  What were the results and what are normal ranges that the authors compared the values for this bird against?

line 133

it would be useful to have a summary of the results indicating the resistance pattern of the Pseudomonas and the reason for the selection of amikacin as opposed to a less toxic antibiotic.  What was the duration of amikacin therapy?

line 134

why was this dose selected?  once a day doses for raptors are listed as 15-20mg/kg SID or twice a day doses 10-20mg/kg twice a day (see recent Carpenter exotic animal formulary 6th edition 2023

line 136

trade name of product please

line 146

using a face mask or intubation?

line 209

what were the reasons for the death of the bird - euthanasia, if so for what reasons?

line 212

 I do not understand what you mean by a falcon flap - is this a description of the lesion?

line 218

I would have thought that if this bird was flown for falconry that the exertion and stretching of the skin as a result of muscular flight would have interfered with the healing of the wounds, which I assume were not sutured but left to heal by second intention?

Author Response

Comments Line 26/Line 48: I would agree with the other reviewers comments - have you any supporting evidence.  This should be available from the many breeding centres involved with breeding falcons commercially and also from the conservation breeding centres.   - I am not sure there is evidence for higher than expected prevalence - this is non-sensical.  you refer to Forbes line 48 who reports prevalence is greater than previously thought...what does this mean...greater than he thought or someone else - this is just an opinion - it would be a valid point if it was greater than a previous study....but greater than what someone thought is a pretty low bar in the scale of scientific evidence?

Response: This were support by literature, I just add (some references).

Comments line 81: more case information here please - sex, age (should be known if captive bred) and management (falconry or breeding bird) and diet if known.

Response: Yes, I just do it.

Comments line 88: what do you mean vertical flight - was this falcon still being flown and stooping even with such severe injuries that are pictured in figures 1 and 2.  

Response: Zoo keepers and veterinarians added environmental enrichment based on the assessed level of pain, and the animal was allowed to fly vertically with chicken decoys whenever it wished.

Comments line 104: I find it surprising that there were no significant abnormalities - it almost seems inconcievable that the white cell count was not affected or out of normal ranges.  What were the results and what are normal ranges that the authors compared the values for this bird against?

Response: Clinical analyses were performed at the park by veterinarians and revealed no abnormalities in the blood count. These were based on the parameters of the machines used and data from the birds.

Comments line 133/134: it would be useful to have a summary of the results indicating the resistance pattern of the Pseudomonas and the reason for the selection of amikacin as opposed to a less toxic antibiotic.  What was the duration of amikacin therapy?/why was this dose selected?  once a day doses for raptors are listed as 15-20mg/kg SID or twice a day doses 10-20mg/kg twice a day (see recent Carpenter exotic animal formulary 6th edition 2023

Response: The antibiotic was selected by the zoo vets based on the resistance levels detected in the antibiogram sent by the reference laboratory. The doses correspond to those prior to the updates.

Comments line 136: trade name of product please

Response: I just do it.

Comments line 146:using a face mask or intubation?

Response: Mask.

Comments line 209: what were the reasons for the death of the bird - euthanasia, if so for what reasons?

Response: The cause of death in the autopsy report was due to intestinal parasitic infection. No signs of metastasis were observed. Park management requested that no reference be made to the parasites.

Comments line 212:  I do not understand what you mean by a falcon flap - is this a description of the lesion?

Response: Yes, the aspect/description after the treatment.

Comment line 218: I would have thought that if this bird was flown for falconry that the exertion and stretching of the skin as a result of muscular flight would have interfered with the healing of the wounds, which I assume were not sutured but left to heal by second intention?

Response: The animal was invited to fly with a chick decoy when it was assessed as pain-free, as an attempt was made to improve environmental enrichment. According to human medicine literature, the degree of pain associated with tumor lesions improves after electrochemotherapy. In this animal's case, it flew of its own accord, according to park caretakers.

Reviewer 3 Report

Comments and Suggestions for Authors

Dear Authors,

The manuscript certainly touches upon an interesting topic. Oncology of birds and mammals is becoming more and more common. It is certainly necessary to combat this and conduct research to identify and treat patients. The authors describe an interesting case. But the manuscript cannot be published in this form. The title of the manuscript should clarify the essence of the study. The text of the manuscript is mixed up in different chapters and should be moved to the appropriate chapters. A chapter on Materials and Methods is needed. The comparative part in the discussion needs to be expanded and additional literature sources on other species should be cited. The article provides too brief a description of them. There is not enough comparison. It is necessary to focus on writing the conclusion of the manuscript based on the study conducted in accordance with the presented hypothesis. It does not exist yet. After all the comments have been corrected, the manuscript can be reviewed again.

Comments on the Quality of English Language

Dear Editor,

The manuscript certainly touches upon an interesting topic. Oncology of birds and mammals is becoming more and more common. It is certainly necessary to combat this and conduct research to identify and treat patients. The authors describe an interesting case. But the manuscript cannot be published in this form. The title of the manuscript should clarify the essence of the study. The text of the manuscript is mixed up in different chapters and should be moved to the appropriate chapters. A chapter on Materials and Methods is needed. The comparative part in the discussion needs to be expanded and additional literature sources on other species should be cited. The article provides too brief a description of them. There is not enough comparison. It is necessary to focus on writing the conclusion of the manuscript based on the study conducted in accordance with the presented hypothesis. It does not exist yet. After all the comments have been corrected, the manuscript can be reviewed again.

Author Response

Comment: The manuscript certainly touches upon an interesting topic. Oncology of birds and mammals is becoming more and more common. It is certainly necessary to combat this and conduct research to identify and treat patients. The authors describe an interesting case. But the manuscript cannot be published in this form. The title of the manuscript should clarify the essence of the study. The text of the manuscript is mixed up in different chapters and should be moved to the appropriate chapters. A chapter on Materials and Methods is needed. The comparative part in the discussion needs to be expanded and additional literature sources on other species should be cited. The article provides too brief a description of them. There is not enough comparison. It is necessary to focus on writing the conclusion of the manuscript based on the study conducted in accordance with the presented hypothesis. It does not exist yet. After all the comments have been corrected, the manuscript can be reviewed again.

Response: I totally agree. I just added data from other therapies to the discussion, relating it to the article. I added the Material and Methods too. 

Round 2

Reviewer 3 Report

Comments and Suggestions for Authors

Dear Editor,

The manuscript has been corrected and additional data have been added. Diagnostics and treatment of avian oncology is of interest and deserves attention in research. But I still believe that the sample in the study is insufficient for conclusions. I leave the right to make a decision on publishing or refusing to publish this message to the editor. There are many different facts in ornithology, but not all are confirmed with a sufficient sample. The article takes into account comments on the methodology. At the same time, the author's theoretical training is at the proper level. The results of previous studies by other authors have been taken into account.